# Gliflozins: From Antidiabetic Drugs to Cornerstone in Heart Failure Therapy—A Boost to Their Utilization and Multidisciplinary Approach in the Management of Heart Failure

**DOI:** 10.3390/jcm12010379

**Published:** 2023-01-03

**Authors:** Lorenzo Pistelli, Francesca Parisi, Michele Correale, Federica Cocuzza, Francesca Campanella, Tommaso de Ferrari, Pasquale Crea, Rosalba De Sarro, Olga La Cognata, Simona Ceratti, Tonino Recupero, Gaetano Ruocco, Alberto Palazzuoli, Egidio Imbalzano, Giuseppe Dattilo

**Affiliations:** 1Department of Biomedical and Dental Sciences and Morphofunctional Imaging, Section of Cardiology, University of Messina, 98122 Messina, Italy; 2Cardiothoracic Department, University Hospital Policlinico Riuniti, 71100 Foggia, Italy; 3Cardiology Unit, Riuniti of Valdichiana Hospitals, USL-SUD-EST Toscana, 53045 Montepulciano, Italy; 4Cardiovascular Diseases Unit, Cardio Thoracic and Vascular Department, S. Maria alle Scotte Hospital, University of Siena, 53018 Siena, Italy

**Keywords:** heart failure, pharmacological therapy, gliflozins, SGLT-2 inhibitors, chronic heart failure

## Abstract

Heart failure (HF) is a complex, multifactorial, progressive clinical condition affecting 64.3 million people worldwide, with a strong impact in terms of morbidity, mortality and public health costs. In the last 50 years, along with a better understanding of HF physiopathology and in agreement with the four main models of HF, many therapeutic options have been developed. Recently, the European Society of Cardiology (ESC) HF guidelines enthusiastically introduced inhibitors of the sodium-glucose cotransporter (SGLT2i) as first line therapy for HF with reduced ejection fraction (HFrEF) in order to reduce hospitalizations and mortality. Despite drugs developed as hypoglycemic agents, data from the EMPA-REG OUTCOME trial encouraged the evaluation of the possible cardiovascular effects, showing SGLT2i beneficial effects on loading conditions, neurohormonal axes, heart cells’ biochemistry and vascular stiffness, determining an improvement of each HF model. We want to give a boost to their use by increasing the knowledge of SGLT2-I and understanding the probable mechanisms of this new class of drugs, highlighting strengths and weaknesses, and providing a brief comment on major trials that made Gliflozins a cornerstone in HF therapy. Finally, aspects that may hinder SGLT2-i widespread utilization among different types of specialists, despite the guidelines’ indications, will be discussed.

## 1. Introduction

HF is an extremely complex, multifactorial, progressive clinical condition, characterized by cardiac function impairment secondary to a structural and/or functional heart abnormality [1]. The natural history of HF is characterized by progressive worsening and deterioration of patient performance status [1,2,3,4,5,6,7,8]. A total of 64.3 million people worldwide are estimated to have HF, with a prevalence of 2% in the general population and up to 12% in people older than 65 years old [4,8]. Up to 5% of all hospital admissions are due to HF, and this number is projected to increase by 50% in the next 25 years with a 30-day re-admission rate of 20–30% [4,8,9]. Ischemic heart disease remains the leading cause of HF in western countries, followed by hypertension, non-ischemic dilatative cardiomyopathy, valvular and congenital heart disease, arrhythmias and degenerative cardiomyopathies [1,10,11,12,13]. Neurohormonal and biochemical dysregulation are main actors in HF progressive worsening and performance status deterioration [1,14,15,16,17]. Other factors sucas endothelial dysfunction, chronic inflammation, oxidative stress and metabolic dysregulation have a critical role in sustaining this condition [1,4,5,6,7,18,19,20,21].

HF has been divided into distinct phenotypes based on the assessment of left ventricular (LV) systolic function, expressed through LV ejection fraction (EF) (LVEF). According to the 2021 European Society of Cardiology (ESC) guidelines for diagnosis and treatment of HF [1], we distinguish: (1) HFrEF in patients with reduced LVEF (LVEF ≤ 40%); (2) HFmrEF in patients with a LVEF between 41% and 49% (mildly reduced LV systolic function); and (3) HFpEF in patients with symptoms and signs of HF, with structural and/or functional cardiac abnormalities and/or raised natriuretic peptides, with an LVEF ≥ 50%.

HFrEF is characterized by a substantial cardiomyocyte loss, resulting in the development of systolic dysfunction. In HFrEF patients, the volume overload is most often the result of permanent neurohumoral activation (RAA-System). 

HFpEF is characterized by structural and cellular alterations leading to an inability of the LV to relax properly, e.g., cardiomyocyte hypertrophy, intercellular fibrosis, altered cardiomyocyte relaxation and inflammation.

Differences in the pathological development of HFrEF and HFpEF have been reported for aspects of inflammation, endothelial function, cardiomyocyte hypertrophy and death and fibrosis [22].

Unlike HFrEF, the HFpEF syndrome is characterized by a systemic proinflammatory state induced by comorbidities (including obesity, diabetes mellitus (DM), chronic obstructive pulmonary disease, renal failure and hypertension) as the main cause of myocardial structural and functional alterations [23]. Systemic inflammation also affects other organs such as the lungs, skeletal muscle, and kidneys, leading, respectively, to pulmonary hypertension, muscle weakness, and sodium retention [24]. 

Sodium glucose cotransporter type 2 inhibitors (SGLT2-i) represent a new class of anti-hyperglycemic agents for type-2 DM (T2DM), which act insulin-independently to selectively inhibit renal glucose reabsorption, thereby increasing urinary glucose excretion [25]. SGLT2i was the first class of glucose-lowering drugs able to reduce cardiovascular events in diabetic and non-diabetic patients with HF [see Trials section]. 

The mechanisms through which SGLT2-i may improve cardiovascular outcomes are not completely understood. Moreover, despite the potential benefits of SGLT2-i in reducing adverse cardiological and nephrological events, SGLT2-i is underprescribed for eligible patients [24]. In order to obtain the maximum use of these drugs, we want to give a boost to their use by increasing the knowledge of the SGLT2-i and better understanding the mechanisms of this new class of drugs, highlighting strengths and weaknesses and providing a brief comment on major trials that made Gliflozins a cornerstone in HF therapy. Finally, aspects that may hinder SGLT2-i large utilization among different types of specialists despite the indications in the guidelines will be discussed.

## 2. Mechanism of Action and Effects of SGLT2-i

The mechanisms by which gliflozines impact mortality and morbidity are not completely understood, particularly on the basis that beneficial cardiovascular effects have been observed irrespective of renal function in patients with reduced glucose filtration and thus glucose excretion [26,27]. Nowadays, the most persuasive and fascinating hypothesis seems to be that gliflozins intervene on different HF models by modifying loading conditions, neurohormonal axes and heart cells’ biochemical and vascular functions, resulting in an improvement of all HF aspects. Below we will summarize the effects of SGLT2i on these models.

### 2.1. Cardio-Renal Model and Modifying Loading Conditions

Considering the critical role of cardiac-induced renal dysfunction in the pathophysiology of HF, it is possible to explain that the observed cardiac benefits may be embedded in modulated renocardiac signaling. Although gliflozins act mostly as diuretics, they, contrary to loop diuretics, have lasting beneficial effects on functional class, rates of HF hospitalization and mortality in patients with HF. Osmotic diuresis and natriuresis induced by SGLT-2i may help maintain the correct load condition, resulting in an improvement in blood volume [28,29]. However, this natriuresis is not associated with neurohormonal activation, potassium loss or renal dysfunction. This favorable diuretic profile provides an important advantage in the management of volume status in patients with HF and may justify the superior long-term HF outcomes. Analysis from the EMPA-REG OUTCOME trial highlighted how change in load condition was the first determinant of CV death reduction in patients treated with empagliflozin [29].

The SGLT2-i nephroprotective effect is also demonstrated by the reduction of albuminuria and macroalbuminuria; these could be linked to increased sodium delivery at the macula densa and afterwards activation of tubuloglomerular feedback, which enhances afferent arteriolar tone and decreases intraglomerular pressure [30].

Moreover, gliflozins seem to also inhibit the sodium-hydrogen exchanger on tubular cells (Figure 1), determining lower sodium and water reuptake [31]. The renal protective action of gliflozins may contribute more to the clinical benefits of gliflozins than their actions on metabolism mitochondria and vascular smooth muscle.

Gliflozins determine osmotic diuresis and natriuresis by inhibiting SGLT-2 and NHE, respectively, on tubular cells. Blocking both SGLT-2 and NHE determines a reduction in sodium re-uptake and, consequently, increased water loss with an improvement in loading conditions. 

SGLT-2: sodium-glucose cotransporter 2; NHE: sodium-hydrogen exchanger.

### 2.2. Neurohormonal Model

The sympathetic nervous system (SNS) regulates glucose metabolism in various organs, including the kidneys. An important crosstalk between the SNS and SGLT2 regulation may not only explain SNS-induced modifications of glucose metabolism but may contribute to the cardiovascular and renal function improvement shown in subjects in therapy with SGLT2-i [32]. In fact, sympathetic inhibition by SGLT-2i can be regulated by the central autonomic system, and this mechanism may show how SGLT-2i improves left ventricle function [33]. Furthermore, beneficial effects of SGLT2-i are directly or indirectly related to a hemodynamic effect due to SGLT2 influence on the RAA system, including the attenuation of intracardiac fibrosis (Figure 2) [34]. A lack of increase in heart rate despite reductions in blood pressure and plasma volume may suggest decreased SNS activity [35]. Compared with classical diuretics, SGLT2-i may have different effects on the interstitial and intravascular compartments; dapagliflozin administration provides greater electrolyte-free water clearance and greater fluid clearance from the interstitial fluid space than from the circulation via peripheral sequestration of osmotically inactive sodium [26]. This may also limit the deleterious effects of reflex neurohumoral stimulation that usually occurs in response to intravascular volume depletion with traditional diuretics [36]. In HF, myocardial remodeling occurs through different pathways, resulting in apoptosis, fibroblast proliferation and fibrosis as a consequence of RAAS overactivation, hyperadrenergic tone and a chronic inflammatory state [37,38]. Sodium-proton exchanger (NHE), a transport protein, is hyperactivated in HF models [39]. NHE would determine increased cytoplasmatic levels of sodium and, consequently, calcium. High calcium cytoplasmatic levels are strongly related to myocardial cell damage and apoptosis. [38] It has been hypothesized that SGLT2i would be able to directly block NHE with a reduction in Na and fluid retention and improve cardiac remodeling through cardiac NHE inhibition and consequent intracellular calcium reduction in favor of mitochondrial calcium [40,41].

### 2.3. Anti-Inflammatory Model

Gliflozins would be further effective on cardiac remodeling because of their anti-inflammatory effects [26,37,42]. In fact, SGLT2i has been proven to reduce circulating inflammatory and pro-fibrotic mediators, particularly IL-6, TGFBeta 1, which is involved in myofibroblast proliferation, and adipokines, and to reduce oxidative stress, allowing a reassertion of the M1/M2 macrophage ratio (Figure 2) [38,42,43]. SGLT2 was demonstrated to promote M2 macrophage activation through oxidative stress reduction and to determine TGF-Beta1 attenuation, resulting in a slowing of fibrous tissue deposition and cardiac remodeling processes [37,42,44,45,46].

SGLT2-i, by reducing levels of leptin (a pro-inflammatory adipokine that appears to have a considerable role in HFpEF onset), would indirectly lower aldosterone levels and, consequently, sodium reuptake and myocardial fibrosis [47]. How SGLT2i lowers leptin levels remains to be understood, and it is not simply justifiable by their effect on adipose tissue reduction [47].

Empaglifozin was associated with a reduction in epicardial fat at 6 months [48]. Epicardial fat may be an important determinant in cardiac dysfunction in light of its strict interaction with the heart. In fact, epicardial adipose tissue (EAT) is a local source of catecholamines and proinflammatory cytokines, leading macrophages to switch to the M1 phenotype [48], and thus predisposes to arrhythmias, progressive cardiomyocyte death, tissue inflammation and cardiac fibrosis and remodeling, differently from pericardial fat, which is relatively more metabolically “inert” [49].

Dapagliflozin is associated with a reduction in both atrial flutter/atrial fibrillation and ventricular arrhythmias, probably through a reduction in EAT [50]. However, reducing EAT may not be beneficial when associated with increased inflammation.

Results derived from the EMPA-TROPISM study [48] showed that empagliflozin significantly improved adiposity, interstitial myocardial fibrosis, aortic stiffness and inflammatory markers in nondiabetic patients with HFrEF. SGLT2i may be anti-inflammatory agents by acting either indirectly by improving metabolism and reducing oxidative stress or by direct modulation of inflammatory signaling pathways [51].

Gliflozins’ beneficial effect on myocardial remodeling. SGLT-2i determines the reduction of IL 6, TGF-Beta, adipokines and aldosterone, promotes reassessment of the M1/M2 macrophage ratio and reduces epicardial fat, resulting in lower apoptosis, oxidative stress, collagen deposition and fibroblast proliferation.

SGLT-2: sodium-glucose cotransporter; TGF-B: transforming growth factor—Beta; M1: M1 macrophage; M2: M2 macrophage; IL6: interleukin-6.

### 2.4. Cardio-Circulatory Model

From the cardio-circulatory model’s point of view, SGLT-2i would be effective in reducing afterload by reducing cardiovascular resistances (Figure 3). NHE hyperactivity leads to higher cytoplasmatic calcium levels with vascular constriction. Lowering of cytoplasmatic calcium at this level would result in vascular relaxation and, consequently, to the drop in arterial stiffness described in previous studies [40,52,53,54]. A possible interaction of dapagliflozin with the K+ voltage channel, a major determinant for resting membrane potential and thus vascular tone, would determine vasodilation [55]. Dapagliflozin was observed to directly determine increased PKG activity, another important factor involved in vasodilation [55]. The impairment in vascular function contributes to the progression of CHF by increasing the afterload. An improvement of vascular function after treatment with empagliflozin was demonstrated in HF patients [56]. Empagliflozin causes a decrease in the stiffness of the aorta and the proximal branches, reducing the afterload of the left ventricle. The decreased afterload of the LV may contribute to the beneficial effects of SGLT2 inhibition. The resulting improvement of ventricular-arterial coupling may contribute to the improved prognosis observed in patients with HF under SGLT2 inhibition.

In an observational, non-randomized study in patients with CHF and T2DM, switching from other oral hypoglycemic drugs to SGLT2i demonstrated improved endothelial function [57]. The direct effects of SGLT2 inhibition on vascular function, combined with the natriuresis effects of SGLT2 inhibition, may contribute to hemodynamic effects [58].

Gliflozins reduce afterload by reducing cardiovascular resistance. SGLT-2i inhibits NHE expressed on endothelial and vascular smooth muscle cells, resulting in a decrease in cytoplasmatic calcium with vascular relaxation and the consequent drop in arterial stiffness. Furthermore, SGLT-2i interacts with the K+ voltage channel on vascular smooth muscle and increases PKG activity, which is involved in vascular tone.

SGLT-2: sodium-glucose cotransporter; PKG: phosphokinase: NHE: sodium-hydrogen exchanger. Kv: K+ voltage channel; Na-Ca E: Na-Ca exchanger.

### 2.5. Biochemical Model

In HF models, a change in the energetic metabolism of the heart has been observed with an increased use of ketones (in particular beta-hydroxybutyrate), which apparently are more oxygen-sparing [59,60]. Ketone bodies supply energy to tissues when glucose is not readily available (e.g., hypoglycemia or insulin resistance) or in the case of exceeding levels of fatty acids, such as in the case of lipolysis activation. In end-stage HF, hyperadrenergic tone determines insulin resistance and increased lipolysis with higher levels of circulating ketones [60].

A switch in the energetic substrate would be an adaptive response of a failing heart to increase its own efficiency. B-hydroxybutyrate is thought to have an oxidative stress scavenging action and to reduce oxidative stress [61]. Moreover, B-hydroxybutyrate would inhibit histone deacetylases in myocytes (Figure 4), suppressing transcription of genes involved in heart hypertrophy and fibrosis [61,62]. According to this fascinating “superfuel” theory, it would be reasonable to suppose that SGLT2i, through a not yet well understood mechanism that involves insulin level reduction and lipolysis, would improve heart efficiency and remodeling by increasing B-hydroxybutyrate levels [59,60,61,63]. This hypothesis would also, in part, possibly explain the “obesity paradox” observed among HF patients [5,6,64].

The increased ketogenesis improves cardiac function and mechanical efficiency [65]. Empagliflozin improves adverse cardiac remodeling in HF by stimulating the switch of myocardial fuel utilization away from glucose toward ketone bodies, improving myocardial energetics, improving systolic function and getting better cardiac reverse remodeling [66].

The results of the EMPA-VISION trial [67], a double-blind, randomized, placebo-controlled, mechanistic study with the aim of determining the effects of empagliflozin treatment on cardiac energy metabolism and physiology using magnetic resonance spectroscopy (MRS) and cardiovascular magnetic resonance (CMR), have not yet been published. However, due to the suspension of face-to-face contact imposed in March 2020 to restrict transmission of COVID-19, the number of patients included in the analysis of efficacy for the HFpEF was substantially reduced, and this cohort was also underpowered for the planned analysis of the primary endpoint.

In the DEFINE-HF (Dapagliflozin Effects on Biomarkers, Symptoms and Functional Status in Patients With HF With Reduced Ejection Fraction) trial [68], in HFrEF patients, dapagliflozin increased ketone-related and short-chain acylcarnitine as well as medium-chain acylcarnitine principal components analysis-defined metabolite clusters compared with placebo. Changes in metabolites were related to changes in Kansas City Cardiomyopathy Questionnaire scores (i.e., worse quality of life) and in NT-proBNP levels, without interaction by treatment group.

Gliflozins improve heart energetics through an increase in B-hydroxybutyrate, which allows greater ATP production at the cost of lower O_2_ consumption. Moreover, B-hydroxybutyrate has oxidative stress scavenging action and inhibits histone deacetylases, suppressing the transcription of genes involved in heart hypertrophy and fibrosis.

ROS: reactive oxygen species; ATP: adenosine triphosphate; O_2_: oxygen.

## 3. Comments to Trials

The EMPA-REG OUTCOME [52] was the first study that gave rise to the hypothesis of an effect of SGLT2i on the cardiovascular outcome: 7028 diabetic patients (BMI < 45 kg/mq, eGFR > 30 mL/min/1.73 mq and HbA1c between 7 and 10) were randomized to receive standard or high dose empagliflozin (respectively, 10 mg or 25 mg) or placebo and followed for a median observation time of 3.1 years. The study proved empagliflozin effectiveness in reducing deaths from cardiovascular causes, hospitalizations for HF and deaths from any cause compared to a placebo and irrespective of the administered dose. Relative risk reduction was respectively 38%, 35% and 32% for cardiovascular mortality (*p* < 0.001), HF hospitalization (*p* = 0.002) and death from any cause (*p* < 0.001) [52]. Subsequent subanalysis highlighted that even in non-HF patients Empagliflozin allowed a decrease in risk for HF hospitalization and cardiovascular death independently from basal HF risk category, proving empagliflozin beneficial effect independently from the presence of HF at baseline [53,69].

A few years later, the EMPEROR-Reduced trial [70] with a randomized group of 3730 patients, NYHA classes II to IV, eGFR > 20 mL/min/1.73mq and ejection fraction below 40% to receive empagliflozin (10 mg once daily) or placebo in order to assess empaglifozin effectiveness in HFrEF, irrespective of the presence of diabetes. Results supported the use of empaglifozin to reduce the total number of hospitalizations for HF (*p* < 0.001) and improve quality of life (QoL), and it proved to be more effective among patients with lower EF. Furthermore, a protective effect of SGLT-2i on the kidney was observed, with a slower decline in the estimated glomerular filtration rate (eGFR) among the empaglifozin group (*p* < 0.001) [70]. These results were confirmed by similar studies [71,72]. In particular, the DAPA-HF trial showed dapaglifozin efficacy in reducing by 26% the relative risk for the combined outcome of HF hospitalization/urgent visits for HF and cardiovascular mortality among HFrEF patients with NYHA classes II-IV and eGFR > 30 mil/min/1.73 mq [71]. EMPEROR-Reduced [70], in comparison to DAPA-HF [71], enrolled patients with a mean lower ejection fraction, higher N-terminal pro BNP levels and lower eGFR (subjects with an eGFR between 20 and 30 mil/min/1.73 mq were also enrolled in the EMPEROR-Reduced trial). An extremely interesting meta-analysis by Zannad et al. [73] assessed a 13% reduction in all-cause death with SGLT-2i (whether empaglifozin or dapaglifozin), with comparable effects of the two drugs in improving outcome and in adverse event rate. The SOLOIST-WHF trial showed that sotagliflozin administration in diabetic patients recently hospitalized for worsening HF reduces cardiovascular mortality, further hospitalizations and urgent visits compared to placebo in both HFrEF and HFpEF patients [74]. However, the early termination of the trial (due to a loss of funding from the sponsor) and the relatively poor number of HFpEF patients enrolled do not allow us to firmly conclude in favor of sotagliflozin’s employment in HFpEF patients [74]. Differently, another SGLT-2i, ertuglifozin, was inexplicably not superior to placebo in improving cardiovascular outcomes among a population that was similar to those analyzed in trials involving empaglifozin, dapaglifozin and sotaglifozin [75]. EMPEROR-Preserved proved empagliflozin effectiveness also in HFpEF patients [70]. However, it should be considered that patients enrolled in the EMPEROR-Preserved trial [76] had more comorbidities and lower EF and higher NT-proBNP levels compared to those in other HFpEF studies such as PARAGON-HF [77,78,79]. More recently, the DELIVER trial, which investigates dapagliflozin effectiveness in HFpEF, showed comparable results to EMPEROR-preserved [76,80]. Interestingly, the data showed that dapagliflozin had a similar effectiveness among patients with LVEF below 60% and above 60% [80]. Notably, both the DELIVER and EMPEROR-Preserved trials were significant for a composite endpoint of hospitalization/unscheduled visit for HF or death for HF, while DAPA-HF and EMPEROR-Reduced showed a significant p value in the reduction of both hospitalization for HF and death for HF [70,71,76,80]. A meta-analysis of the most important trials on gliflozins and HF (DELIVER, EMPEROR-Preserved, EMPEROR-Reduced, DAPA-HF and SOLIST-WHF) further emphasized the positive effect of SGLT2-I among the entire HF spectrum, irrespective of the EF, confirming their role as a cornerstone in HF therapy [81].

All the above-mentioned studies investigated gliflozin’s effects on stable, ambulatory patients. Differently, a recent trial, EMPA-RESPONSE-AHF, further validated by the EMPULSE trial, randomized subjects hospitalized for HF to receive empagliflozin or a placebo within 24 h from the admission [82,83]. The primary outcome from the EMPULSE trial was the composite of death from any cause, HF events and time to further events and QoL. A total of 530 patients with acute de novo or decompensated chronic HF were randomized to empagliflozin 10 mg once daily or placebo, irrespective of LVEF, and a reduction in all-cause death and improvement in QoL (*p* = 0.0054) were shown for patients treated with empagliflozin [82]. Thanks to these results and in consideration of the fact that SGLT2-i showed no significant interaction with other HF drugs [84], a recent document from the European HF Working Group encourages the introduction of empagliflozin as soon as possible as a first-line therapy in HF patients in order to achieve the maximally tolerated optimal medical therapy in the shortest time. The clinical benefits of gliflozins are unquestionable in HF patients. In contrast, the mechanisms of action of gliflozins remain disputed. At the international level, there is agreement on recognizing a gap between extremely positive clinical trials and uncertain mechanisms of action.

## 4. How to Overcome Doubts about Prescribing Gliflozins

According to the most recent ESC HF Guidelines [1], empagliflozin or dapagliflozin are recommended (class I A) in HFrEF patients to reduce the risk of HF hospitalization and death. Beyond the beneficial effects that have been previously described, it is important to underline that SGLT-2i does not need up-titration. This is relevant information, considering that most HF drugs need titration in order to provide a larger benefit.

The reluctance to prescribe gliflozins may be linked to cost, and national drug agencies for reimbursement of SGLT2i require prescribers to fill out treatment plans, effectively delaying the start of treatment with SGLT2i. Like angiotensin receptor-neprilysin inhibitors, gliflozins may not be as beneficial when the syndrome of HF progresses.

Moreover, SGLT-2i side effects are very minor, making gliflozins extremely well tolerated drugs.

Most frequent side effects are urogenital tract infections, more often fungal infection, which have been described to occur more likely among women and the elderly [85,86,87]. Urogenital infection may potentially worsen to pyelonephritis and urosepsis.

More rare side effects are hypoglycemia, excessive volume depletion, Fournier’s gangrene and ketoacidosis, while the possibilities that gliflozins affect cholesterol levels and favor bone fracture levels are still being debated. Hypoglycemia episodes are rare and verified exclusively when another anti-diabetic drug is administered with gliflozins [88,89]. A total of 0.3% to 4.4% of patients may present volume depletion symptoms, which may complicate hypotension and dehydration; thus, special attention should be paid in order to avoid excessive volume depletion [89]. Hypotension risk is more consistent in patients who have already been treated with loop diuretics and ARNI [1]. Between 2013 and 2015, twelve cases of Fournier’s gangrene secondary to SGLT-2i administration were reported, and no cases were mentioned among the EMPA-REG OUTCOME trial population [69,84]. About cholesterol levels, a rise in LDL cholesterol have been described, but whether or not Gliflozins determine an increase in LDL-cholesterol is a question still being debated since trials showed discordant results [52,85,86]. On the other hand, the increase in HDL levels in these patients was well demonstrated [52,85,86]. Another debated topic is bone fractures. The CANVAS Trial data supported the hypothesis of a possible association between Canagliflozin and both bone fracture and lower limb amputation [90]. However, results are not conclusive, and a relation between a higher rate of lower limb amputation and bone fractures has never been proven [82]. Similarly, despite the fact that some data suggested a higher cancer risk associated with SGLT2i therapy, subsequent analyses and studies confuted any relation [89]. Contrarily, it is well known that, as a consequence of their lipolytic effect, gliflozins increase ketones in the blood and thus may lead to ketoacidosis. The Food and Drug Administration (FDA) reported approximately 2500 cases of diabetic ketoacidosis from 2014 to 2016, more frequently among insulin-treated patients and even in the presence of glucose blood levels below 250 mg/dl [90,91,92,93]. In the perioperative setting, this issue becomes serious. According to 2020 FDA advisory warnings [92] for increased incidence of euglycemic diabetic ketoacidosis, when SGLT2i medications are continued prior to non-cardiac surgeries and the patients are referred for surgery, it is important to advise patients to stop canagliflozin, dapagliflozin and empagliflozin for at least 3 days and ertugliflozin for at least 4 days before surgery because of the increased risk of perioperative euglycemic ketoacidosis in diabetic patients. Even if the optimal duration of interruption is still debated and no evidence is available about when these drugs should be discontinued before surgery in HF patients. In these patients, discontinuation of the drug may be deleterious to HF management [93].

With the exception of urogenital tract infections, which are estimated to happen in up to 8% of patients receiving gliflozins, all the above-mentioned side effects are extremely rare [52,85,86]. What could truly hinder different types of specialists from an extremely wide use of these drugs in a HFrEF setting is probably renal failure. A higher risk of acute kidney injury in patients undergoing SGLT-2i has not been proven. Even if volume depletion may potentially lead to kidney deterioration, afferent arteriolar constriction, which is induced by SGLT-2i, reduces intraglomerular pressure, slowing glomerulosclerosis [28]. As mentioned before, major trials tested gliflozin effects on patients with renal function impairment (eGFR > 20–25 mL/min/1.73 mq), and their safety was proved even among those patients. What is truly affected by renal function impairment is gliflozin’s effect on lowering blood glucose, as a lower GFR means lower glucose filtered through the glomerulus and thus lower glucose urinary excretion. Moreover, a nephroprotective effect was observed: subsequent studies showed how the eGFR reduction per year was attenuated by dapagliflozin independently from the presence of HF, suggesting an independent nephroprotective action of SGLT-2i [52,70,72,94]. Thus, physicians should not consider chronic kidney disease (CKD) an absolute contraindication to their administration, and gliflozin use among patients with CKD and an eGFR > 25 mL/min/1.73 mq should be encouraged rather than discouraged. This appears even more relevant if we consider epidemiological data about HF and CKD: 83% of outpatients with HF are estimated to have impaired renal function [95]. In line with these considerations, it is important to underline that SLT2i has been proposed to improve even erythropoietin secretion, improving anemic status, which is typical of CKD patients and affects about 25–30% of HF patients, further worsening the condition [3]. However, it is correct to underline that anemia in HF has a multifactorial origin: either cardio-renal, malabsorptive and secondary to a chronic inflammatory status [2].

Notably, HF should be considered a complex syndrome with relevant implications for diabetes and CKD; they worsen each other in a vicious cycle that determines poor outcomes [96]. Gliflozins may improve either renal function either glucose blood levels and cardiac function and remodeling, representing the ideal drug to interrupt the circles or slow their progression.

Otherwise from what previously asserted by Brauwald, more recently it has been suggested that sodium-glucose transport blockage would attenuate transport in the proximal tubular segments, leading to a different oxygenation of the medullary and cortical kidney, resulting in a protective action on the renal cortex, but, conversely, damaging medullary structures [50,97]. Nevertheless, a positive net effect is desirable. Data from the EMPA-Kidney trial [98], which enrolled more than 6000 patients with CKD (mean eGFR of 37.5), will be helpful in shedding further light on gliflozin’s exact role in renal disease; its results are expected to be in line with those from the CREDENCE trial and DAPA-CKD [94,99]. Notably, cardiologists should remember that a modest, transitory rise in serum creatinine during the initial phase of gliflozin therapy is possible; however, it does not represent an indication to interrupt therapy. In such cases, strict monitoring of renal function is the most appropriate option [1].

## 5. Conclusions

Despite the fact that some SGLT2i mechanisms are not yet fully understood, their impact on outcomes, quality of life and both laboratory and echocardiographic parameters in HF patients is brilliant and involves several pathophysiological mechanisms. All these beneficial effects come at the cost of a well-tolerated drug with very few side effects and no need for titration. Contraindications are few, and CKD could become a contraindication only for eGFR < 25 mL/min/1.73 mq, but, contrarily, the therapy with SGLT2i should be encouraged in HF patients for the nephroprotective effect.

## Figures and Tables

**Figure 1 jcm-12-00379-f001:**
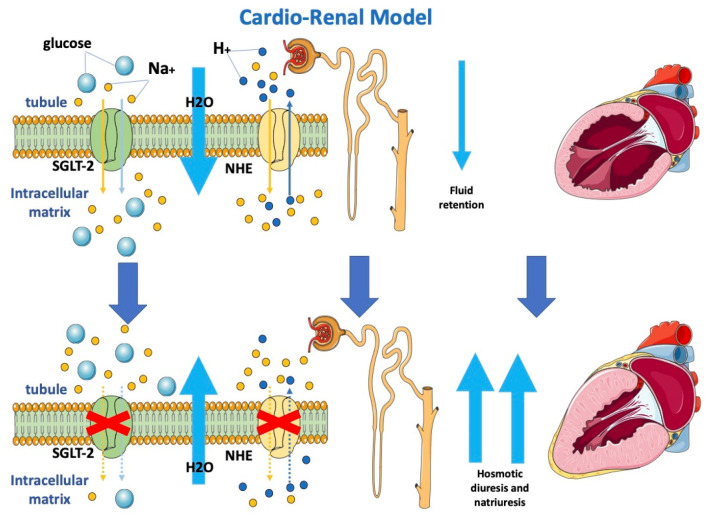
Effects of SGLT2-i on the cardio-renal model.

**Figure 2 jcm-12-00379-f002:**
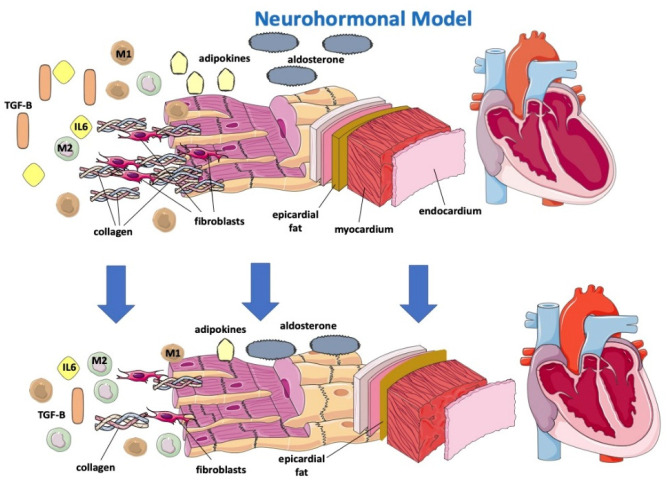
Effects of SGLT2-i on a neurohormonal model and on inflammatory markers.

**Figure 3 jcm-12-00379-f003:**
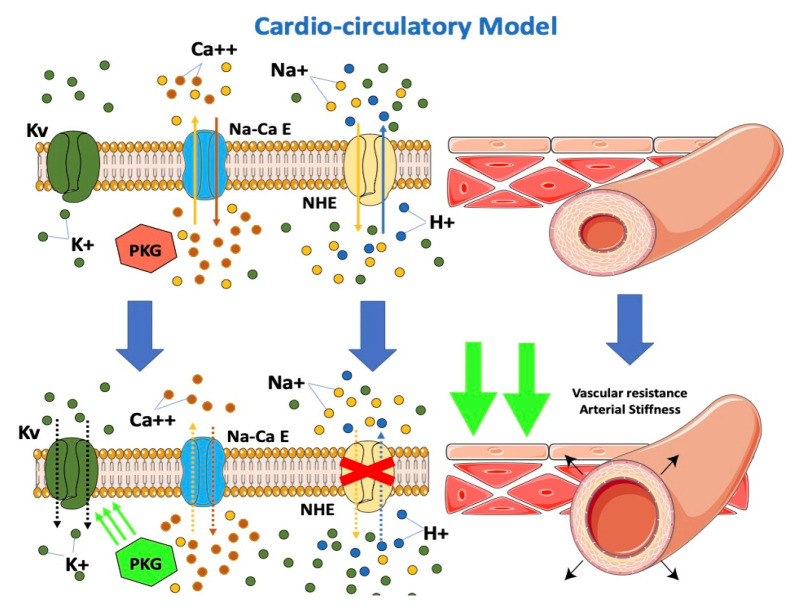
Effects of SGLT2-i on the cardio-circulatory model.

**Figure 4 jcm-12-00379-f004:**
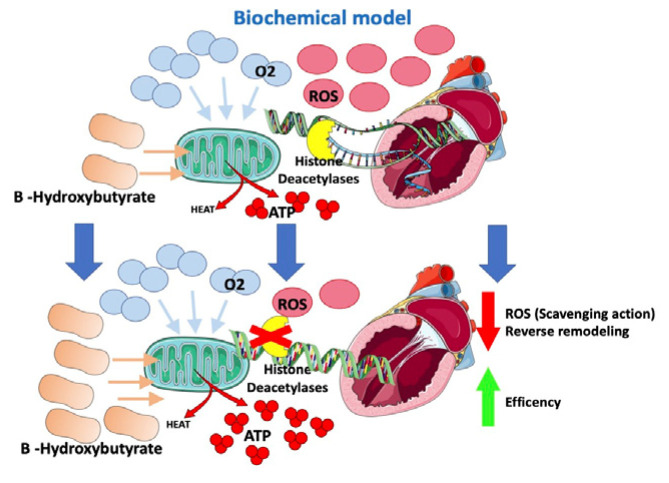
Effects of SGLT2-i on the biochemical model.

## Data Availability

Not applicable.

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
