# Peer review of "Gliflozins: From Antidiabetic Drugs to Cornerstone in Heart Failure Therapy—A Boost to Their Utilization and Multidisciplinary Approach in the Management of Heart Failure"

_jcm, 2023, doi:10.3390/jcm12010379_

Round 1

Reviewer 1 Report

I have read with interest the review article by Pistelli et al entitled “Gliflozins: from antidiabetic drugs to cornerstone in Heart Failure therapy. A boost to their utilization and multidisciplinary approach in the management of heart failure “

In this review article the authors give a broad overview of the possible mechanisms of action of SGLT2-i as angular treatment of patients with heart failure with reduced ejection fraction.

The review is very useful as it is a current issue. The review is extensive and carried out with great rigor. Contrasted mechanisms of action are explained and hypothesized about other mechanisms that are not so clear. Finally, a review of the main clinical trials that include iSGLT2 in heart failure with both depressed and preserved systolic function is carried out.

As the only comment, define the initials EAT on line 160 since it is the first time they appear in the text.

Author Response

Reviewer #1:

I have read with interest the review article by Pistelli et al entitled “Gliflozins: from antidiabetic drugs to cornerstone in Heart Failure therapy. A boost to their utilization and multidisciplinary approach in the management of heart failure“. In this review article the authors give a broad overview of the possible mechanisms of action of SGLT2-i as angular treatment of patients with heart failure with reduced ejection fraction.

The review is very useful as it is a current issue. The review is extensive and carried out with great rigor. Contrasted mechanisms of action are explained and hypothesized about other mechanisms that are not so clear. Finally, a review of the main clinical trials that include iSGLT2 in heart failure with both depressed and preserved systolic function is carried out.

As the only comment, define the initials EAT on line 160 since it is the first time they appear in the text.

Response: We would thank our reviewer for comments and suggestion. We defined the initials EAT in the text.

Reviewer 2 Report

I received for review an article prepared by Lorenzo Pistelli et al. entitled "Gliflozins: from antidiabetic drugs to cornerstone in Heart Failure therapy. A boost to their utilization and multidisciplinary approach in the management of heart failure", which is being processed for publication in the Journal of Clinical Medicine (IF=4,964).

The introduction is laconic. The aim of the work has not been precisely described anywhere. Too much attention has been paid to information at the elementary level, such as the renal glucose threshold and the mechanism of action of gliflozins. I do not see any justification for describing the indications in which phlorizin was used, a substance that is a prototype for gliflozins. The presentation of such information suggests a lack of any specific vision for the structure and purpose of the work, which results in too much attention being paid to elementary information that adds little to the topic of the work. At work, information from basic research and information from clinical trials are intertwined in an unstructured manner. The various mechanisms to explain the beneficial effects of gliflozins in heart failure are described very chaotically and briefly.

Author Response

Reviewer #2

I received for review an article prepared by Lorenzo Pistelli et al. entitled "Gliflozins: from antidiabetic drugs to cornerstone in Heart Failure therapy. A boost to their utilization and multidisciplinary approach in the management of heart failure", which is being processed for publication in the Journal of Clinical Medicine (IF=4,964).

  • The introduction is laconic.

Response: The introduction was changed according to reviewer’s suggestion.

  • The aim of the work has not been precisely described anywhere.

Response: At the end of the introduction we have explained better the purpose of the paper.

  • Too much attention has been paid to information at the elementary level, such as the renal glucose threshold and the mechanism of action of gliflozins.

Response: The authors deleted information at the elementary level.

  • I do not see any justification for describing the indications in which phlorizin was used, a substance that is a prototype for gliflozins.

Response: We have deleted it.

  • The presentation of such information suggests a lack of any specific vision for the structure and purpose of the work, which results in too much attention being paid to elementary information that adds little to the topic of the work.

Response: We have revised the text according to reviewer’s suggestion, deleting elementary information.

  • At work, information from basic research and information from clinical trials are intertwined in an unstructured manner.

Response: We have better organized the text, trying to divide the information coming from basic research from that derived from clinical trials. 

  • The various mechanisms to explain the beneficial effects of gliflozins in heart failure are described very chaotically and briefly.
    Response: We have better explained possible mechanisms that justify beneficial effects of gliflozins in heart failure.

Reviewer 3 Report

Dear authors, thank you for this sound and complete review over  inhibitors of the Sodium glucose cotransporters as therapy for HF. 

I only have one recommendation to make: In line 310-312 you are discussing the occurence of ketoacidosis during the course of Gliflozins. However, in the perioperative setting this issue becomes more and more important. Several national guidelines recommend actually to stop Glifozins 3 days before major cardiac and non-cardiac surgery due to severe ketoacidosis in the post-operative setting. Please add a statement about this fact in your revision.

Author Response

Reviewer #3

Dear authors, thank you for this sound and complete review over  inhibitors of the Sodium glucose cotransporters as therapy for HF. 

I only have one recommendation to make: In line 310-312 you are discussing the occurence of ketoacidosis during the course of Gliflozins. However, in the perioperative setting this issue becomes more and more important. Several national guidelines recommend actually to stop Glifozins 3 days before major cardiac and non-cardiac surgery due to severe ketoacidosis in the post-operative setting. Please add a statement about this fact in your revision.

Response: We would thank our reviewer for comments and suggestion. We have added this statement.

Round 2

Reviewer 2 Report

I received for review a revised version of the article prepared by Lorenzo Pistelli et al. entitled "Gliflozins: from antidiabetic drugs to cornerstone in Heart Failure therapy. A boost to their utilization and multidisciplinary approach in the management of heart failure", which is being processed for publication in the Journal of Clinical Medicine (IF=4,964). The effort of the Authors to improve the work should be appreciated. However, despite the fact that the work has a certain substantive value, its level differs from the level of works published in journals with Impact Factor.  I present my most important arguments in favor of defending this thesis below.

1.     The introduction of the work is long, but not really informative. Heart failure is a very complex issue. In the introduction, the Authors presented elementary information on the pathogenesis of heart failure, but did not refer to the classification of heart failure in terms of the assessment of left ventricular systolic function and did not indicate differences in the pathogenesis of heart failure with reduced and preserved systolic function. The presented information is fragmentary and superficial. On the other hand, a lot of space is devoted to the outline of the history of changing views on the pharmacological treatment of heart failure. This seems redundant, and what is more, the authors did not show why these dominant views of heart failure pharmacotherapy changed. The fragment of the introduction devoted to changes in views on the treatment of heart failure gives the impression of a popular science text rather than a scientific one.

2.     I stand by my opinion expressed in the first review that an entire chapter on the mechanism of action and the history of gliflozins is completely unnecessary. The most important information presented here should be included in the introduction. Currently, gliflozins are widely used drugs. So there is no need to explain their mechanism of action in such detail, because it is an elementary knowledge.

3.     I'm having a hard time getting to the third chapter. Actually, I don't fully understand what its purpose is.

4.     The fourth chapter does not actually bring any new information in relation to the elementary knowledge about the mechanism of action of gliflozins, which results in a diuretic and natriuretic effect.

5.     In the following chapters, I believe, very little is shown how the use of SGLT2 inhibitors has a beneficial effect in heart failure, from the perspective of the discussed mechanisms.

6.     The text is not prepared very carefully. Many of the abbreviations used were not explained in the text when first used.

7.     The list of references was not prepared in accordance with the rules required by the MDPI.

In conclusion, the collected material can be used, rewritten and submitted again, apart from the current submission.

Author Response

Reviewer #2:

I received for review a revised version of the article prepared by Lorenzo Pistelli et al. entitled "Gliflozins: from antidiabetic drugs to cornerstone in Heart Failure therapy. A boost to their utilization and multidisciplinary approach in the management of heart failure", which is being processed for publication in the Journal of Clinical Medicine (IF=4,964). The effort of the Authors to improve the work should be appreciated. However, despite the fact that the work has a certain substantive value, its level differs from the level of works published in journals with Impact Factor.  I present my most important arguments in favor of defending this thesis below.

  1. The introduction of the work is long, but not really informative. Heart failure is a very complex issue. In the introduction, the Authors presented elementary information on the pathogenesis of heart failure, but did not refer to the classification of heart failure in terms of the assessment of left ventricular systolic function and did not indicate differences in the pathogenesis of heart failure with reduced and preserved systolic function. The presented information is fragmentary and superficial. On the other hand, a lot of space is devoted to the outline of the history of changing views on the pharmacological treatment of heart failure. This seems redundant, and what is more, the authors did not show why these dominant views of heart failure pharmacotherapy changed. The fragment of the introduction devoted to changes in views on the treatment of heart failure gives the impression of a popular science text rather than a scientific one.
  2. Response: We would thank our reviewer for comments and suggestion. We added classification of heart failure in terms of the assessment of left ventricular systolic function and reported differences in the pathogenesis of heart failure with reduced and preserved systolic function. Furthermore, in order to short the introduction we deleted the part on the history of pharmacological treatment of heart failure.

  1. I stand by my opinion expressed in the first review that an entire chapter on the mechanism of action and the history of gliflozins is completely unnecessary. The most important information presented here should be included in the introduction. Currently, gliflozins are widely used drugs. So there is no need to explain their mechanism of action in such detail, because it is an elementary knowledge.

Response: We would thank our reviewer for comments and suggestion. The most important information presented in the chapter of mechanism of action and the history of gliflozins were included in the introduction, deleting the old second chapter.

  1. I'm having a hard time getting to the third chapter. Actually, I don't fully understand what its purpose is.

Response: Third chapter is not real isolated chapter but it is only the presentation of the effects of glyflozines on the different models of heart failure, which will be presented later. As a simple presentation of what will be described next, it has been reduced as an extension and redefined second chapter.

  1. The fourth chapter does not actually bring any new information in relation to the elementary knowledge about the mechanism of action of gliflozins, which results in a diuretic and natriuretic effect.

Response: fourth chapter is redefined as 2.1. The authors changed all chapters according reviewer suggestion.  

  1. In the following chapters, I believe, very little is shown how the use of SGLT2 inhibitors has a beneficial effect in heart failure, from the perspective of the discussed mechanisms.

Response: in these chapters the authors have added and highlighted the parts that may indicate a clinical benefit of SGLT2i. The following chapters are redefined 2.2, 2.3

  1. The text is not prepared very carefully. Many of the abbreviations used were not explained in the text when first used.

Response: We would thank our reviewer for suggestion. The text was checked and abbreviations described when first used.

  1. The list of references was not prepared in accordance with the rules required by the MDPI.

Response: the references were changed according rules requiring by the MDPI.

In conclusion, the collected material can be used, rewritten and submitted again, apart from the current submission.

Yours sincerely,

Michele Correale, MD, PhD, MSc, Fellow Italian Society of Cardiology. Policlinico Riuniti University Hospital, Foggia, Italy